# A Framework for Exploring Player Perceptions of LLM-Generated Dialogue in Commercial Video Games

**Nader Akoury**
University of Massachusetts Amherst
nsa@cs.umass.edu

**Qian Yang**
Cornell University
qianyang@cornell.edu

**Mohit Iyyer**
University of Massachusetts Amherst
miyyer@cs.umass.edu

## Abstract

The growing capabilities of large language models (LLMs) have inspired recent efforts to integrate LLM-generated dialogue into video games. However, evaluation remains a major challenge: how do we assess the player experience in a commercial game augmented with LLM-generated dialogue? To explore this question, we introduce a dynamic evaluation framework for the dialogue management systems that govern the task-oriented dialogue often found in roleplaying video games. We first extract dialogue from the widely-acclaimed role-playing game *Disco Elysium: The Final Cut*, which contains 1.1M words of dialogue spread across a complex graph of utterances where node reachability depends on game state (e.g., whether a certain item is held). Using this dataset, we have GPT-4 perform *dialogue infilling* to generate grounded utterances based on game state represented via code. In a statistically robust study of 28 players recruited from the `r/DiscoElysium` subreddit, the LLM outputs are evaluated against the game designers' writing via both preference judgments and free-form feedback using a web interface that recreates the game's core conversation functionality. Overall, the game designers' prose is significantly preferred to GPT-4 generations, with participants citing reasons such as improved logical flow and grounding with the game state. To spur more principled future research in this area, we release our web interface and tools to enable researchers to build upon our work.[1]

## 1 Introduction

Dialogue in most narrative-driven video games has historically been static: players may choose from a small number of *pre-written* dialogue options that depend on the game's state (e.g., items held or goals achieved). The advent of large language models (LLMs) has inspired efforts to dynamically *generate* dialogue in video game environments, such as

those by AI Dungeon (Walton, 2019), InWorld AI (Gelfenbeyn et al., 2021), and ConvAI (Mukherjee, 2022), which can potentially imbue games with endless variety.[2] However, evaluating the impacts of LLM-generated dialogue on the *player experience* has yet to be tackled in a principled manner.

In this paper, we directly evaluate the player experience by asking video gamers to interact with LLM-generated dialogue injected into *Disco Elysium: The Final Cut*, a highly-acclaimed dialogue-centered video game (Kurvitz et al., 2021).[3] To mitigate difficulties with evaluating open-ended text generation (Karpinska et al., 2021), we specifically examine a *constrained* dialogue generation task in which an LLM must decide how to update a dialogue to match the corresponding game state.

Take for example a scene in which the game's protagonist (an amnesiac detective) is interrogating an uncooperative suspect. If the player decides to act like the suspect's friend to obtain more information, the variable `seafort.deserter_sugg_you_are_buddies` is set to **true**, and the corresponding line of dialogue is:

> *You can tell **me, here**. It won't be \*that\* usable.*

Continuing the example, we then prompt an LLM to appropriately modify the dialogue when the variable `seafort.deserter_i_am_also_communist` is set to **true**. We evaluate the generated dialogue against the game writers' original line:

> *You can tell **a comrade**. It won't be \*that\* usable.*

While the semantics of the utterance remains mostly unchanged, the player's assumed communist persona is reflected in the second dialogue. These dialogue options, and any associated non-player character (NPC) responses, are defined us-

---

[1] https://pl.aiwright.dev

[2] See for example Nvidia's recent ACE demo.

[3] *Disco Elysium: The Final Cut* is currently rated the #1 PC video game of all time on Metacritic.

ing a graph structure commonly referred to as a "dialogue tree" in the video game industry. Crucially, our task setup does not expect an LLM to generate all of the game's dialogue (as in e.g., AI Dungeon), but rather provides the LLM with human-written dialogue as input and tweaks it to fit various dynamic aspects of the game state. To condition on the game state, we devise a clever approach of encoding the graph structure as a mix of code and natural language, which also opens the possibility in future work of modifying the game state in response to a generated utterance. This constrained setup makes the task tractable to evaluate and also practically relevant to future LLM-human collaborative game writing applications.

Can an LLM understand enough about the game state to appropriately modify dialogue in a way that is logically and tonally consistent with the game state? More generally, how does LLM-generated dialogue stack up with the award-winning dialogue written by *Disco Elysium*'s designers, and what do video gamers think are the biggest issues with it? By choosing a popular roleplaying game with a dedicated following, we are more easily able to find participants familiar with the expected tone and lore needed to effectively assess our generated dialogue. We evaluate OpenAI's state-of-the-art GPT-4 LLM (OpenAI, 2023) via a statistically robust user study, asking *Disco Elysium* fans to provide preference judgments and free-form feedback within an interface designed to mimic the game's dialogue engine.

Perhaps unsurprisingly, players strongly prefer the original dialogue (H) compared to LLM-generated dialogue (G), with participants citing reasons such as better logical consistency (H: 61% vs G: 28%) and flow (H: 67% vs G: 21%). However, participants note that GPT-4 begins to close the gap at providing interesting dialogue options (H: 47% vs G: 36%) that advance their goals (H: 57% vs G: 33%), though further work is required to ground the dialogue to the game state as 32% of generations rated by at least one player are deemed illogical upon reading the next utterance. To facilitate future research in player-centered video game dialogue generation, we release our annotation interface and tools to reproduce our dataset.

## 2 Related Work

Commercial video games have become increasingly popular testbeds for neural approaches to grounded language (Suhr et al., 2019) and reinforcement learning (Bellemare et al., 2012; Kempka et al., 2016). To the best of our knowledge, only the sandbox game Minecraft has been explored as a testbed for interactive dialogue research (Volum et al., 2022), despite some commercial video game dialogue being explored in non-interactive settings (Lopez Latouche et al., 2023; Weir et al., 2023).

Dialogue systems in roleplaying video games are task-oriented (Grosz, 1974), where quests in the game act as tasks that the player must complete within the constraints of the game world. The static pre-written dialogue graphs are a form of finite-state dialogue management (Brabra et al., 2022). Using such a rigidly constrained dialogue management approach ensures the authorial intent of the game writers at the expense of more natural conversation flows. Rather than upend these familiar techniques for more flexible approaches which have yet to gain traction (Riedl and Young, 2004; Mateas and Stern, 2005), likely due to their complexity, we opt to augment the existing approaches actively in use in commericial video games. Our work bridges the tightly scripted scenarios common to video games, with more natural speech that offers humans more agency over their interaction with virtual agents, leading to easier to design agent interactions useful for training simulations (Demasi et al., 2020) and tutoring (Wang et al., 2023a).

Narrative-driven video games fall under the umbrella of interactive storytelling, which can take on many forms including tabletop roleplaying games like Dungeons and Dragons (Callison-Burch et al., 2022), choose your own adventure books (Clark and Smith, 2021), and interactive fiction (Hausknecht et al., 2020). More broadly, graph representations have been used for finite-state dialogue management (Koller et al., 2018), and as knowledge bases (Gritta et al., 2021) for slot-filling approaches (Cohen, 2019) to task-oriented dialogue.

## 3 The *Disco Elysium* Dataset

In *Disco Elysium: The Final Cut*, the player takes on the role of a down-on-his-luck detective in order to solve a murder mystery in a dystopian city. The majority of in-game interactions are in the form of dialogue, including interactions with not only other characters but also inanimate objects (e.g., a ceiling fan and a bathroom mirror from the first scene of the game), which makes the game well-suited for

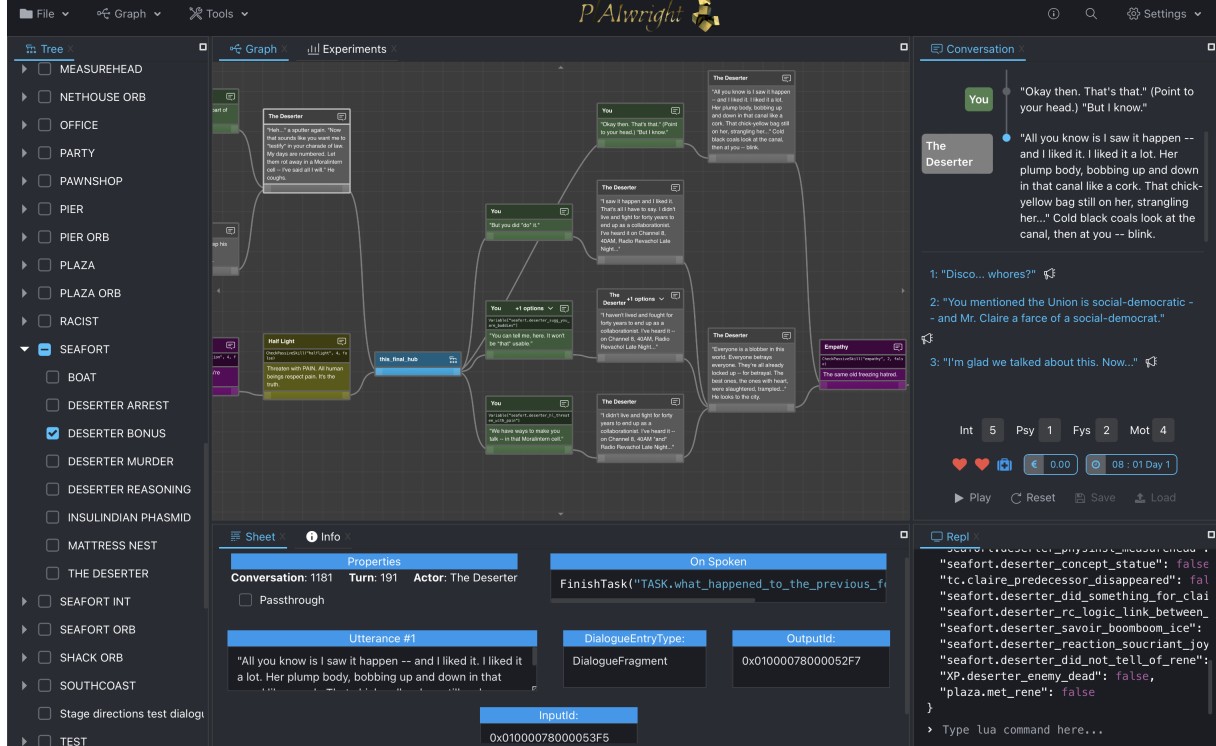

Figure 1: As part of the effort to decipher the data format from *Disco Elysium: The Final Cut*, we created a tool to display and analyze the dialogue graphs from the game. This tool allows for filtering data in various ways, displaying attributes of the dialogue nodes, visualizing entire conversations, creating dataset splits (Section 3.1; Appendix A), preprocessing dialogues into Lua scripts (Section 4.2), analyzing experiments (Section 6), and more.

our experiments. Additionally, the game's state is encoded in Lua[4] via descriptively-named boolean variables and getter/setter functions that trigger on certain nodes of the dialogue graph. In this section, we describe how we extract and process the rich dialogue graph (Figure 1) and game state from *Disco Elysium* for our constrained dialogue generation task.

### 3.1 Extracting data from *Disco Elysium*

We begin by extracting a catalog of all top-level entities (characters, items, conversations, dialogue entries, and game state variables) from a purchased PC version of *Disco Elysium: The Final Cut* using the open source tool AssetStudio.[5] Its prose, with over 70K utterances consisting of roughly 1.1M words of dialogue (Table 1), is nearly twice the length of *Atlas Shrugged*. As with many games, *Disco Elysium* encodes game state variables and functions into Lua expressions that are run by the game engine based on the player's actions.

---

[4] https://www.lua.org
[5] https://github.com/Perfare/AssetStudio

### Dataset Splits

| | Train | Valid | Test | Total |
|---|---|---|---|---|
| | 89.8% | 5.4% | 4.7% | |
| Utterances | 65,316 | 4,143 | 3,237 | 72,696 |
| Words | 1,001,191 | 59,877 | 52,816 | 1,113,884 |
| Nodes | 98,442 | 6,950 | 5,092 | 110,484 |
| Forks | 17,283 | 1,544 | 964 | 19,791 |
| Variables | 99,015 | 6,989 | 5,132 | 111,136 |

Table 1: We split the data into training, validation, and test sets based on the number of dialogue words, while ensuring an approximately similar proportion of conditional dialogue forks and referenced Lua variables. See Table A1 for more information.

**Dialogue and game state encoding:** The game state of *Disco Elysium* is an exhaustive mix of variables and functions that are descriptively named and commented. All dialogue entries in the graph include metadata about the character who is speaking, as well as any preconditions (boolean-valued expressions) required to speak the utterance. For example, the first dialogue option in Figure 2's Lua

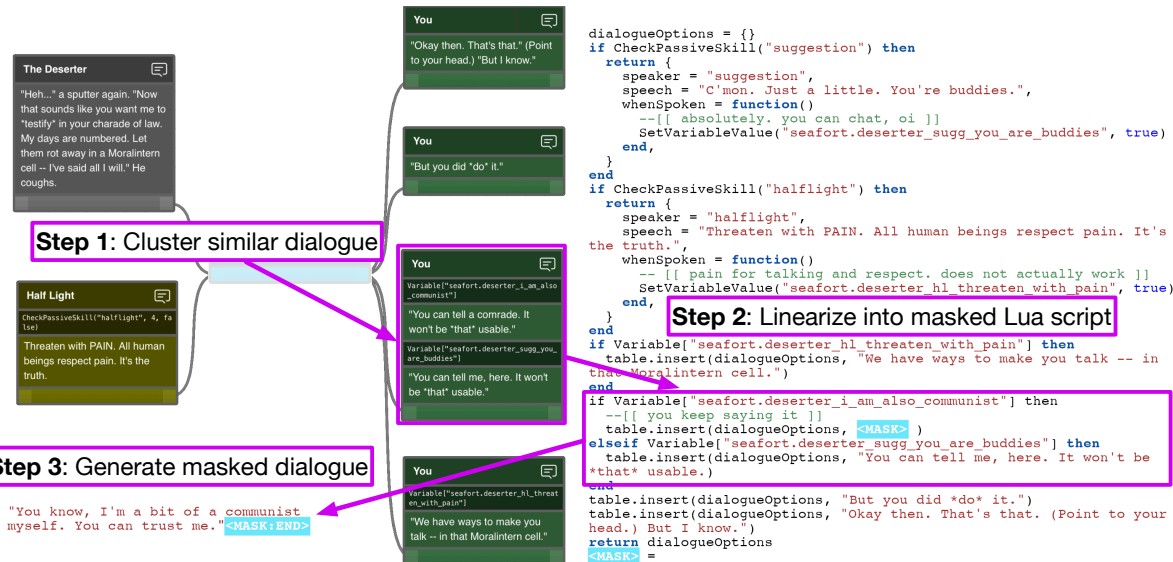

Figure 2: In this conversation from *Disco Elysium: The Final Cut*, we first cluster dialogue nodes by semantic similarity of game state variables and spoken dialogue. Then, we linearize the next turn of dialogue in the graph into a Lua script that contains game logic and dialogue. Finally, we <MASK> one utterance from the cluster and ask GPT-4 (an LLM trained on code and natural language) to infill the masked dialogue.

script contains the following precondition:

```lua
if CheckPassiveSkill("suggestion")
```

We also observe comments written by the game developers used to document the intent of a variable or function, as in this comment before setting the variable `seafort.deserter_hl_threaten_with_pain`:

```lua
pain for talking and respect. does not
actually work
```

These comments provide a rich, natural source of context to better explain the vast array of entities (Table A1) referenced throughout the game logic. Each dialogue entry may also contain functions that can alter game state when uttered that also serve as important context to understand the dialogue, as in:

```lua
SetVariableValue(
    "seafort.deserter_sugg_you_are_buddies",
    true
)
```

**Creating dataset splits:** Since the dialogue in *Disco Elysium* ultimately tells a single overarching story, it is not possible to create a dataset split in which each fold contains a disjoint set of characters, items, and game state variables. The game's dialogue graph implicitly forms a hypergraph, with hyperedges defined by the Lua variables. Optimal partitioning of a hypergraph is known to be NP-hard, and an exhaustive enumeration is often

more efficient for smaller hypergraphs than specialized algorithms (Papa and Markov, 2007). We use the branch and bound algorithm to enumerate all valid partitions of the dialogue graph that satisfy an $\epsilon = 1.5\%$ variation from the desired splits of 90%/5%/5% train/valid/test.[6] The final split is achieved with minimal overlap in game variables (Table 1).[7]

## 4 Grounded dialogue infilling

As prior text generation and dialogue research has clearly demonstrated (Karpinska et al., 2021; Clark et al., 2021), evaluating LLM-generated dialogue is a daunting challenge. We examine a more constrained subtask in which an LLM is given multiple lexically-similar human-written responses to a given utterance, each of which is slightly different from each other based on the game state. One of these responses is masked out, and an LLM is asked to generate it based on cues from the game state (e.g., that communists have historically referred to one another as comrade). While this task is strictly easier than open-ended dialogue generation,

---

[6]Luckily, many distinct conversations are connected by one or more dialogue edges, making it such that a small handful of connected components in the graph make up roughly 70% of the dialogue and thus are required to be in the training set.

[7]While the rest of this paper uses the validation set, including as a source of few-shot demonstrations to prompt GPT-4, we describe preliminary fine-tuning experiments on the training set in Appendix C.

we find that state-of-the-art LLMs still struggle to solve it. This section details the data filtering and preprocessing steps we performed, as well as the few-shot prompting strategy we use with GPT-4.

## 4.1 Clustering lexically-similar utterances

We detect lexically-similar utterances by applying a simple token-based clustering algorithm on the nodes in the dialogue graph. Starting from a source dialogue node, we traverse all outgoing directed paths which terminate upon encountering a dialogue node; we collect all such sets by using each dialogue node as a source. Then, we tokenize the dialogue in each node set on whitespace and punctuation boundaries while preserving common contractions (e.g., *'ll*, *'s*, etc). Next, we compute bag-of-words F1 among utterances within each set to measure similarity, and also apply the same procedure to the associated Lua conditions. Finally, we consider all disjoint subsets for which $F1 >= 0.5$ (for utterances or conditions), which we qualitatively validated as producing clusters of high lexical similarity.[8]

## 4.2 Linearizing clusters into Lua scripts

Now that we have a set of clusters, we convert each cluster into a single Lua script (see Figure 2, right) that can be fed to a language model. We prefix all the characters, items, comments, and variables referenced by the clustered nodes at the top of the script, along with default values and metadata. Each node in the cluster is visited in sequential order, and its Lua conditions, dialogue, and any associated post-speech game state altering actions are included in the script. Lastly, we enumerate all variants of each script by masking out each instance in a cluster one-by-one. Our masked infilling prompts use a *prefix-suffix-mask* (Donahue et al., 2020; Bavarian et al., 2022) ordering, where the *prefix* contains the portion of the script before the masked utterance, followed by the text `<MASK>`; the *suffix* contains the remainder of the script, followed by `<MASK>` =; and the *mask* contains the utterance we want the model to infill, followed by `<MASK:END>`.

## 4.3 Few-shot prompting to infill dialogue

From our set of linearized Lua scripts, we build few-shot prompts to perform dialogue infilling using GPT-4. We first prefix each prompt with the

**Few-Shot Prompt Statistics**

|  | Min | Avg | Max |
| --- | --- | --- | --- |
| Example Length | 158 | 330 | 2228 |
| Examples per Prompt | 15 | 33 | 45 |

Table 2: Here we report statistics for token lengths of each example, along with the number of examples per prompt.

following instruction to guide the model to generate dialogue constrained to the game state:

> You are a creative game designer writing engaging dialogue for a roleplaying game. For each self-contained dialogue script fill in the <MASK> with interesting dialogue using only facts from the script.

Then for a given script, we select demonstrations from our validation set that do not contain any utterances in common with the target script. We fill the full 8k context of GPT-4 with demonstrations, which we qualitatively find to produce utterances that best match the game's writing style (Table 2).[9]

## 5 Setting up a strong user study

Evaluating LLM-generated dialogue within a large commercial video game is a complex undertaking. Both automatic and crowdsourced evaluation lead to misleading and unreliable conclusions for creative generation tasks (Karpinska et al., 2021; Wang et al., 2023b), which motivates expert annotation (Xu et al., 2023; Karpinska and Iyyer, 2023). More importantly, we want to collect evaluations from people who are actually invested in *Disco Elysium*, as a primary goal of our study is to characterize how LLM-generated dialogue affects the *player* experience. This entails collecting evaluations within an interactive setting rather than having annotators rate utterances in isolation. In this section, we specify our evaluation setup, which involves (1) designing an interface that mimics the *Disco Elysium* dialogue engine; (2) conducting pilot tasks to determine common error categories and further refine the interface to reduce annotator burden; and (3) recruiting participants from Reddit who have completed one or more playthroughs of *Disco Elysium* to complete our main user study.

---

[8]See Appendix B for more details on alternate clustering algorithms that we experimented with.

[9]Additional details, including our full prompt template, can be found in Appendix D.

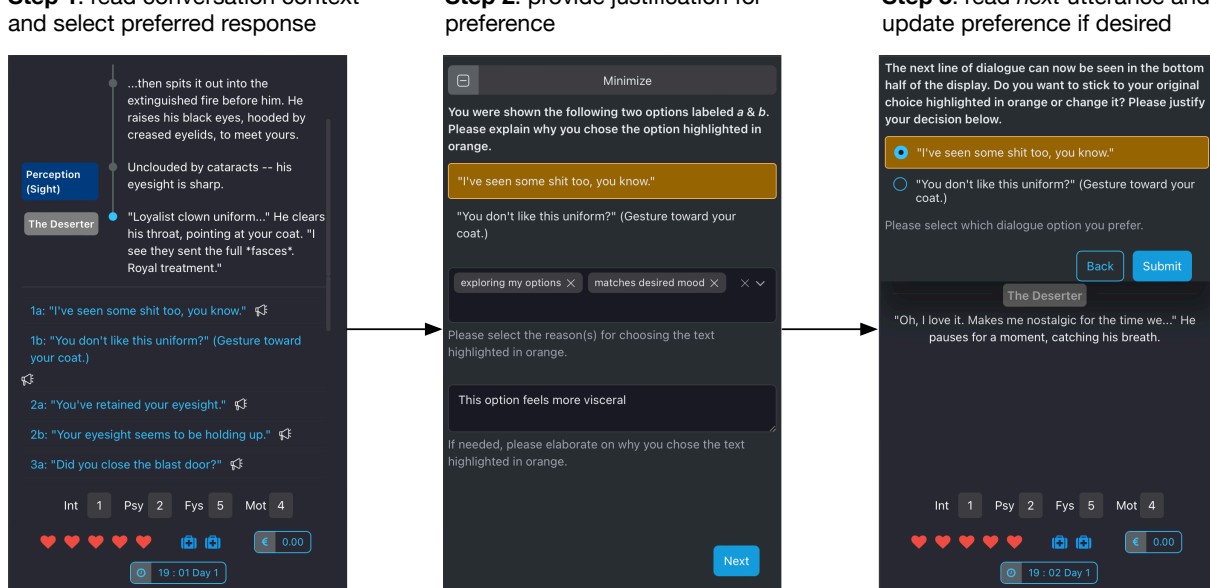

**Step 1**: read conversation context and select preferred response

**Step 2**: provide justification for preference

**Step 3**: read *next* utterance and update preference if desired

Figure 3: Three panels from the mobile version of our web app, which reproduces *Disco Elysium*'s dialogue engine and allows us to collect preference judgments and justifications. In the first panel, players take part in a conversation and are given multiple candidate responses to choose from. Options denoted with *a* and *b* are randomly shuffled between human and LLM-generated dialogue. Upon selecting a paired dialogue option, the second panel is displayed to collect a justification for their choice. The final panel allows players to see the next line of dialogue that will be spoken and optionally change (and justify) their preference.

## 5.1 Designing an interface to recreate *Disco Elysium*'s dialogue engine

In *Disco Elysium*, players control a virtual representation of the main character that can move around and interact with the environment, including initiating conversations. This freedom makes it difficult for us to constrain our study within the confines of the video game. Thus, rather than creating a game mod that includes LLM-generated dialogue,[10] we design a custom web app[11] that recreates the core gameplay systems that underpin the game's dialogue system. This interface allows us to present a specific conversation taken from our validation split to players (i.e., study participants).

**What annotations do we collect?** Players are presented with the original human-written dialogue for all characters from the game, while the main character's dialogue options are paired alongside generated utterances from GPT-4 (Section 4), which we randomly shuffle and label with the subscripts *a* and *b*. Players are then asked to provide a *preference judgment* over the candidate utterances:

choose which candidate best fits their goals while taking into account the previous conversation history and story context (Figure 3a). After making a preference judgment, they are asked to justify their choice via both predefined tags (e.g., **advances my goals** or **matches desired mood**) as well as optional free-form comments (Figure 3b). Finally, they are shown the ground-truth human-written *next* line of dialogue, and they are asked whether they would change their judgment in retrospect given this knowledge; if so, they are again asked to justify their decision (Figure 3c).

## 5.2 Running usability studies to refine the annotation task

We conduct two usability studies to better understand common types of free-form participant feedback; additionally, these studies guide the refinement of our interface to reduce cognitive load for participants (e.g., switching to the two-stage annotation flow presented in Figure 3b & c). These usability studies also led to the coding and integration of predefined tags discussed above. Our first usability study enlists six college students (each of whom had previously played through *Disco Elysium*) to spend one hour with our web app. We also conducted a follow-up controlled observational study

---

[10]See for example the InWorld AI-based Skyrim mod.

[11]We use the React framework with an integrated Lua interpreter that executes the gameplay logic as defined in our linearized Lua scripts (Section 4.2).

**Reddit Player Demographics**

|  | Min | Avg | Max |
|---|---|---|---|
| Hours Played | 25 | 82.6 | 230 |
| Playthroughs | 1 | 2.7 | 11 |

Table 3: The number of hours and playthroughs of *Disco Elysium* as reported by the Redditors who took part in the study.

**Reddit Study Details**

| | |
|---|---|
| Number of Players | 28 |
| Total Utterances Rated | 1,158 |
| Avg Utterances Rated per Player | 41 |
| Avg Utterances Seen per Player | 203 |
| Unique Utterances Rated | 112 |

Table 4: Study details, including number of annotators, the number of unique generated utterances that were rated, and the total number of ratings.

with two more college students, using the Nielsen Norman Group Observer Guidelines[12] to assess the user experience implications of our interface.[13]

**Manually coding human feedback:** We manually code the human-written feedback from our first usability study to build a list of common justifications for player preferences. This leads to a categorization of 13 high-level justifications of a player's initial preference, and 8 reasons for retroactively updating their preference. To improve annotator efficiency, we update our annotation interface to provide a list of common justification tags participants can choose from in addition to free-form text.[14]

### 5.3 Statistically robust Reddit study

Our task necessitates a study with many players since post-utterance functions can alter the current game state, potentially affecting reachability in the graph, which leads to dozens of paths through the conversation chosen from our validation set that we use for our evaluation. For that reason we recruit 28 fans of the game from the r/DiscoElysium subreddit to take part in our study (Table 3).[15] Based on our usability studies, we estimated players require between 1-2 hours to complete our study, and we provide $25 gift cards for participation in the study.

**All participants have played *Disco Elysium* before:** We limit participation in our study to players who have completed at least one full playthrough of *Disco Elysium: The Final Cut* in English.[16] This is necessary since the game weaves a complex narrative that incorporates the player's

dialogue choices. For example, if a player often chooses dialogue options that indicate the main character is a fascist, the player will more frequently be presented with dialogue options that reflect this world view. Thus, each player will have a unique trajectory through the dialogue graph and must mentally keep track of their choices, a skill players learn through experience with the game. Furthermore, due to the dataset split (Section 3.1), our validation data is from a late stage in the game, so participants must know the full story context to understand the nuances of each dialogue choice.

**Study parameters:** Approximately 20% of utterances a player reads during our study contain LLM-generated dialogue. This is another reason why we hired so many participants, as it requires substantial reading time between annotations. In aggregate, 112 unique utterances are rated by our participants, though since each player performs a unique walk of the dialogue graph, on average each player rates 41 utterances, and only 100 utterances receive at least three ratings (Table 4).[17]

## 6 Results & analysis

Overall, our study reveals a strong preference for human-written dialogue over LLM-generated dialogue. Participants most commonly cite reasons for their preferences such as increased appropriateness, better match with their gameplay goals, and stylistic properties (Figure 4). Because assessing the generative capabilities of LLMs is confounded by the subjective nature of rating narrative quality (Ethayarajh and Jurafsky, 2022; Wang et al., 2023b), we also conduct a fine-grained analysis of free-form player justifications to uncover where GPT-4 succeeds and where it needs improvement.

---

[12] https://www.nngroup.com/articles/observer-guidelines/

[13] See Appendix E for more details about our usability studies and refinements.

[14] See Appendix F for specifics on each tag.

[15] Our study was approved by IRB review, and all participants are at least 18 years of age.

[16] We leave evaluation of other languages to future work.

[17] Following Card et al. (2020), our study design has a statistical power of 0.96 for a margin of 10%, though our reported margins in Section 6.1 are often much larger.

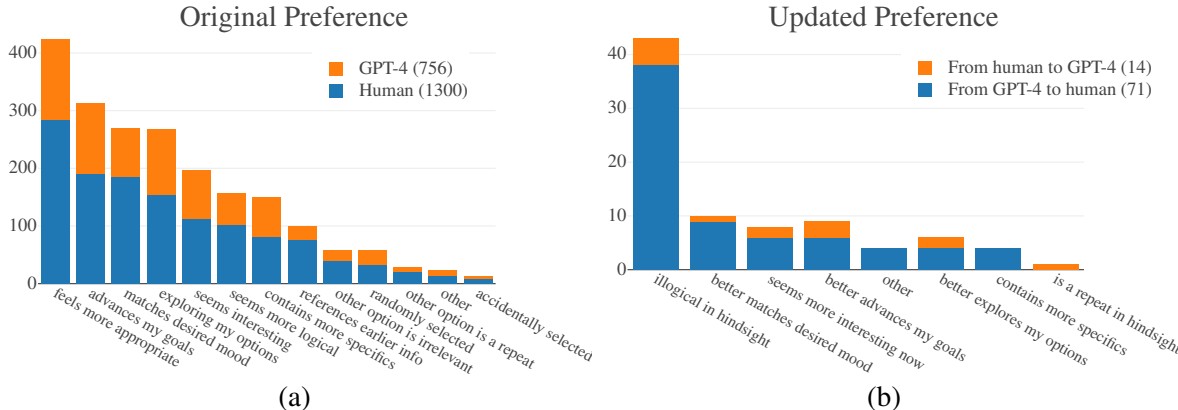

Figure 4: Histogram of tags, aggregated across all 1,158 judgements, (a) upon initially choosing the dialogue, and (b) after a player retroactively updates their preference upon seeing the next utterance in the conversation.

## 6.1 Participants prefer human-written dialogue

Overall, out of the 1,158 total judgements we collect from players, 702 (61%) state a preference for human-written dialogue while 456 prefer GPT-4. When aggregating at the instance level (i.e., computing the majority vote on all instances for which we collected annotations from at least three different players), we note a stronger preference for human-written dialogues. Specifically, upon their first assessment (Figure 3b), players prefer human-written dialogue to that of GPT-4 (H: 64% vs G: 23%; rest ties), and after retrospectively updating their preference (Figure 3c), annotators prefer the original dialogue even more (H: 66% vs G: 23%).

**Reasons for preference:** When players prefer human-written dialogue over model-generated dialogue, they cite reasons such as increased logical consistency (H: 61% vs G: 28%) and flow (H: 67% vs G: 21%). After seeing the next utterance, the most common reason for players to change their preference is that the selected utterance was illogical in hindsight (Figure 4b), which affects 32% of GPT-4 generations rated by at least one player. Overall, these results suggest that future research should focus on better grounding of LLM generations to the game state. However, GPT-4 does close the gap on certain aspects of the generated dialogue, including providing interesting dialogue options (H: 47% vs G: 36%) that contain more specifics (H: 43% vs G: 46%) and advance player goals (H: 57% vs G: 33%), which shows the potential of collaborative human-LLM dialogue.

## 6.2 Fine-grained analysis

While the overall results show a strong preference for human-written dialogue, we note that the task is inherently subjective, and player preference is not always related to the *quality* of the options. Some level of disagreement between participants is expected, which motivates us to perform a more fine-grained analysis to uncover common facets that provide insight on these disagreements and highlight where GPT-4 excels and struggles.

**Inconsistencies with the game world:** Many of the justifications for players' preferences mention appropriateness or logical consistency / flow with the conversation history. In general, the world created by the game designers has a depth and consistency that is difficult for models to understand, especially when it is encoded programmatically (e.g., in Lua scripts) rather than in unstructured text. In the following example, four players justify their preference (one shown below) for the human-written text by noting that the generated utterance does not conform to the game's notion of an "ultra-liberal" character, which requires an understanding of the game's various political factions.

(1) a. *That's a \*choice\*. You could have become self-employed. Create the system.*
HUMAN-WRITTEN

   b. *I steal from the rich, redistribute wealth, and fight for a borderless world.*
GPT-4 GENERATED

     i. As an Ultraliberal character, this option feels more appropriate in this playthrough.
**feels more appropriate**
PREFERS HUMAN-WRITTEN

**Awkward articulation:** Sometimes, the competing objectives of generating a fluent utterance and

staying faithful to the game state result in awkward generations that can impact logical and stylistic coherence, as in:

(2)  a.  ***We're** not 'Coalition-appointed.' We just try to help people.*
HUMAN-WRITTEN

   b.  ***I'm** not 'Coalition-appointed.' We just try to help people.*
GPT-4 GENERATED

   i.  Since Kim is with me, it's more appropriate to say "we".
feels more appropriate, matches desired mood
PREFERS HUMAN-WRITTEN

   ii.  I never know if we can trust Kim
feels more appropriate, matches desired mood,
advances my goals, contains more specifics
PREFERS GPT-4 GENERATED

Here, six players prefer the original utterance, in which both sentences use the plural pronoun "we", and call out the awkwardness of excluding the main character's partner Kim in the second GPT-4 generated sentence; one example justification is shown in (2i). Despite this conflict, however, two players prefer the GPT-4 generation as they believe it better matches the mood (2ii).

**Fit with play style:**  An attractive aspect of role-playing games is that players can mold the game's narrative to their individual play style. *Disco Elysium* allows for a huge variety of play styles; for example, players can choose to tackle the game as an analytic Sherlock Holmes-type detective or as a physically imposing but dim-witted enforcer. GPT-4 is able to provide diverse dialogue that is amenable with certain play styles:

(3)  a.  [Pick up the gun lying in the sand.]
HUMAN-WRITTEN

   b.  *Mind if I examine your gun, Mr. Dros?*
GPT-4 GENERATED

   i.  He threw it away "like an amputated limb" so I don't think asking him for permission rhetorical or not seems appropriate
feels more appropriate
PREFERS HUMAN-WRITTEN

   ii.  I am being passive aggressive here, I do not, in fact, care if he minds.
feels more appropriate
PREFERS GPT-4 GENERATED

In the above example, one player thinks it does not make sense to ask permission to look at the gun (3i), while the other prefers the GPT-4 generation because they intentionally want to be passive aggressive towards Mr. Dros (3ii). Both players marked **feels more appropriate** as a justification, which is not a contradiction since they each have different play styles and objectives.

**Paraphrasing:**  Dialogue in games tends to be static: speaking with a non-player character often leads to the same utterances being repeated in the absence of relevant game state changes. Fortunately, recent LLMs like GPT-4 are quite adept at paraphrasing text. When the model correctly reproduces an utterance semantically similar to the human written dialogue, players often randomly choose between the two:

(4)  a.  *One more time: what have you used this gun for?*
HUMAN-WRITTEN

   b.  *Alright, I'll ask again. What have you been using this gun for?*
GPT-4 GENERATED

   i.  These are both basically the same so I just picked one at random
randomly selected
PREFERS HUMAN-WRITTEN

However, in some cases the generated paraphrases include small extraneous information that feel off to the players. In the following example, two players specifically call out GPT-4's phrasing:

(5)  a.  *Stop changing the subject – we have the murder weapon.* (Point to it.)
HUMAN-WRITTEN

   b.  *Enough squirming. I have the murder weapon, and Kim here can confirm it.*
GPT-4 GENERATED

   i.  I don't think Kim is a figure of authority at this point - he knows about as much about the murder weapon as Harry does.
feels more appropriate,
advances my goals, seems more logical
PREFERS HUMAN-WRITTEN

# 7  Conclusion

In this paper, we perform a user study of LLM-generated dialogue integrated into video games, hiring fans of *Disco Elysium: The Final Cut* to provide fine-grained insights about issues in this domain. We examine a constrained dialogue generation task, in which the game state is integrated into the LLM prompt via Lua scripts that encode game variables, functions, and dialogue. We develop a web interface that reproduces *Disco Elysium*'s dialogue engine to conduct the evaluation. Human-written dialogue is strongly preferred over GPT-4 generations for reasons such as improved logical flow, appropriateness, and tonal consistency. Future work can build on our framework to consider user play style, faithfulness to the game state, and dynamically updating game state as important components of the modeling and evaluation process.

## 8 Acknowledgements

Thanks to all of the *Disco Elysium* fans who took part in the studies described in the paper. Also, thanks to the reviewers and to Brendan O'Connor for valuable feedback which helped improve the manuscript. Finally, thanks to Ronan Salz and George Wei who conducted an early exploration of the *Disco Elysium* data. This project was partially supported by awards IIS-2202506 and IIS-2046248 from the National Science Foundation (NSF).

## 9 Limitations

While the dialogue for *Disco Elysium* is available in Simplified Chinese, Traditional Chinese, English, French, German, Japanese, Korean, Polish, Portuguese-Brazilian, Russian, Spanish, and Turkish, our study focuses exclusively on the English version of the game, which was necessary due to the large number of players required for adequate coverage of the dialogue graph. Additionally, the generated dialogue we show participants is generated once from GPT-4, such that each player rates the same generated utterance. This reduces variation in the annotation task allowing us to have high statistical power for our results. Though due to the static nature of the generated dialogue we do not account for the player's unique walk of the dialogue graph, which would require dynamically generated utterances, even though this could better reflect the interactive nature of video games. We live this to future work. We also note that *Disco Elysium* represents a niche genre of narrative-driven games that does not reflect the full diversity of narratives seen in video games, thus our approach is unlikely to generalize to the diverse catalog of video game narratives. Though our work does apply to a large class of popular video games such as the recently released *Baldur's Gate 3* and *Starfield*[18], both of which have enthralled millions of gamers within a month of their release[19].

## 10 Ethical Considerations

*Disco Elysium* contains adult themes, including discussions of suicide, murder, and rape. For this reason, we ensure participants in our study are at least 18 years of age as required by our institutional review board. We do not collect any demographic information of the participants beyond age verification. Additionally, we ensure participants in our study are fairly compensated for their time, offering $25 gift cards for Reddit participants, $30 gift cards for our observational study participants, and either $25 or $50 gift cards for our initial pilot participants depending on the length of time they spent on the study.

We also note that *Disco Elysium: The Final Cut* is a copyrighted game, so we take special care to ensure we respect the intellectual property of the game's designers.[20] We do not release any models trained on the game's data, nor do we widely release our web interface, which requires registering an account with an authorization token to take part in the study. Finally, the web app only has access to a small portion of the overall game data taken from the validation set which is needed to conduct the study, and that data is only ever kept in memory, never persisted to disk.

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

## A  Dataset Splits

We provide additional details regarding our dataset splits. As we note in the main body of the paper, a key challenge in splitting the *Disco Elysium* dataset into train, valid, and test splits is the high degree of interconnectedness across conversations in the game. While it is not possible to create a dialogue split with a disjoint set of game state variables, we minimize overlaps amongst the splits.

| Variable Overlap | | Dataset Totals | |
|---|---|---|---|
| Train $\bigcap$ Valid | 2897 | Items | 259 |
| Train $\bigcap$ Test | 2871 | Characters | 424 |
| Valid $\bigcap$ Test | 303 | Conversations | 610 |
| (a) | | (b) | |

Table A1: For the *Disco Elysium* dataset: (a) we take special care to minimize the number of referenced variable overlaps amongst the splits; (b) though we do not attempt to disentangle Characters and Items across splits.

## B  Additional Clustering Details

We experiment with a number of algorithms for clustering the game's dialogue, including the Levenshtein distance, Jaccard index, and the Dice coefficient. We also vary the features used for clustering by splitting the words into characters, grouping by ngrams, and through the use of lowercasing. We conduct a manual inspection of the various approaches to clustering, including a hyperparameter sweep of the similarity threshold. This inspection indicated that solely clustering based on the dialogue utterances would either systematically miss semantically similar text, or cluster dissimilar utterances when the similarity threshold was made more permissive.

To combat this tendency, we additionally tried clustering nodes by inspecting the associated Lua conditions. We first parse the Lua expression, extract identifiers (which often refer to functions) and string literals (which often refer to variables). We then split the literals into their constituent words (e.g., `whirling.dreamone_brave` becomes `whirling`, `dreamone`, `brave`), before running the above battery of clustering approaches. In the end, we find that clustering based on a combination of dialogue and Lua expressions produced the best results, without the need for the extra feature engineering, while only relying on the simple Dice coefficient with a threshold $d >= 0.5$.

## C  Preliminary Experiments

We conduct experiments using two LLMs: GPT-3 Curie and Codex (Table A2). GPT-3 Curie is a strong generation model for natural language (Brown et al., 2020), especially when finetuned on a downstream task, while Codex is an extremely capable few-shot LM for code (Chen et al., 2021). As our task contains elements of both natural language and code, it is important to assess the capabilities of each model paradigm.

| Model Class | Prompt Tokens | Model Type | OpenAI API Name |
|---|---|---|---|
| Curie [1] | 2048 | Finetuned | curie |
| Codex | 8000 | Few-Shot | code-davinci-002 |

Table A2: Details of the models used in our experiments. As OpenAI does not provide parameter counts or details on finetuning, we also provide the API name for the models to help reproducibility.

[1]Likely 6.7B parameters, see:
https://blog.eleuther.ai/gpt3-model-sizes/

Since the two models perform different tokenization[2] and support different context lengths, we filter the clusters, keeping only those that fit the smallest context length (2048 tokens) using the GPT-3 tokenizer. We then generate all the linearized scripts representing semantically related text for the next turn of dialogue. After filtering and generating masked variants of the clusters, we are left with 30,501 training examples and 2,668 validation examples.

We finetune Curie for 1 epoch, with a batch size of 32 examples and a learning rate of $0.2\times$ the learning rate of the pretrained model and we weight the loss for the prompt tokens by 0.01. For the few-shot Codex model, we prefix each linearized Lua script with several samples from the validation set such that they take up nearly the full context window (we reserve 100 tokens of the context for generation). We also ensure there are no overlaps in dialogue between the few-shot examples and the script. Consequently, each Codex script has 7 few-shot examples on average.

We choose to measure the performance of the models on the validation set using a bag-of-words F1, as the clustered utterances have a large overlap with the masked text the model is tasked with

[2]Codex uses a modified tokenizer that collapses whitespace since it is commonly used in code formatting.

| Model | Examples | Tokens | BLEURT | F1 |
|-------|----------|--------|--------|-----|
| Curie | 2,668 | 3,041,299 | 41.9 | 25.6 |
| Codex | 2,668 | 21,077,200 | 44.2 | 29.5 |

Table A3: Preliminary experiments over the validation set show that few-shot Codex outperforms a finetuned Curie model for generating context-aware dialogue.

infilling. In addition, we use BLEURT which has proven to be robust for semantic similarity of generated text (Karpinska et al., 2022). Both metrics favor Codex slightly, though given the low F1 score, it's clear the models have much room for improvement on this simplified form of our task. That is to say, naïvely applying our preliminary approach to all the dialogue in the game, not just to the subset of dialogue clustered via similarity, is even more likely to fail. We also posit that Codex likely outperforms Curie since it is a larger model that is explicitly trained on a large corpus of code, even though it uses a few-shot approach to inference.

To better understand the performance difference between the two models we also conduct a small analysis of each model's output. We find that both models tend to copy from the prompt (Table A4), but Codex does it nearly twice as often.

| Model | Examples | Copied |
|-------|----------|--------|
| Curie | 2,668 | 235 |
| Codex | 2,668 | 455 (8)[†] |

Table A4: We find that both Curie and Codex occasionally copy dialogue from the prompt, and in 8[†] instances Codex directly copies a completion from the few-shot examples.

A qualitative inspection of the generations from the Codex model (our best performer) seem to indicate the model may struggle to generate plausible completions due to a lack of historical context to the current conversation. Our script-based prompts do not include any previous dialogue utterances, but rather rely only on the combination of dialogue that can be emitted next and conditional game logic gating those options. It is clear the models also do not make effective use of the game designer's annotations to fill in the gaps. While these comments are likely useful reference for the writers of the game, they may not contain enough context alone to guide generation. Considering Codex has a very long context window and performs better than a finetuned Curie (Table A3), future experiments could attempt

**Few-Shot Prompt Statistics**

| | Min | Avg | Max |
|---|-----|-----|-----|
| Example Length | 158 | 330 | 2228 |
| Examples per Prompt | 15 | 33 | 45 |
| Prompt Length | 7547 | 7566 | 8089 |

Table A5: Here we report statistics for token lengths of each example and the overall prompts, along with the number of examples per prompt.

to include previous turns of dialogue in the prompt to see if that improves generation quality.

## D Few-shot Prompting

Since we target GPT-4 for our main study, we provide few shot prompts using their chat format. All prompts are prefixed with the following system message:

> You are a creative game designer writing engaging dialogue for a roleplaying game. For each self-contained dialogue script fill in the <MASK> with interesting dialogue using only facts from the script.

Additionally, our few shot examples are encoded as chat conversations where a *user* message provides the model with a script, and the the *assistant* responds with the completion. In terms of the linearization we describe in Section 4.2, the *user* message consists of the *prefix* and *suffix*, while the *assistant* message consists of the *mask*. Note we, experimented with other prompting approaches, but found the above worked best. We investigated zero-shot and few-shot prompts containing various numbers of examples. The resultant GPT-4 generations often did not match the style of the game writing, frequently leading to verbose utterances. We also tried interleaving instructions before each few-shot example, and that seemed to have no noticeable difference is quality.

## E Interface adjustments from usability studies

**Two-step annotation flow** In our initial interface players are presented with a single screen in which to provide free-form feedback on both the selected utterance and whether they want to change their mind after seeing the next utterance. This unified feedback screen led players to conflate the reason they chose a dialogue option with their post-hoc reasoning after seeing the next utterance. Before

| Predefined Tag | Description |
|---|---|
| `randomly selected` | You randomly selected between the paired options. |
| `accidentally selected` | You accidentally selected the dialogue option. |
| `advances my goals` | The selected dialogue option advances your goals. |
| `exploring my options` | You are just trying to explore all the dialogue options. |
| `feels more appropriate` | The selected dialogue option fits the conversation better than it's counterpart. |
| `matches desired mood` | The selected dialogue option matches the mood you are going for. |
| `contains more specifics` | The selected dialogue option contains more context specific information than it's counterpart. |
| `other option is a repeat` | The paired dialogue option that was not selected repeats something that was already stated. |
| `other option is irrelevant` | The paired dialogue option that was not select is irrelevant to the current context. |
| `references earlier info` | The selected dialogue option references information from earlier in the conversation. |
| `seems interesting` | The selected dialogue option seems more interesting than it's counterpart. |
| `seems more logical` | The selected dialogue option fits the current context more logically than it's counterpart. |
| `other` | Please explain in your own words why you selected the dialogue option. |

Table A6: List of predefined tags and their associated description from our web app for justifying why a player *initially chose a candidate utterance*.

our observational study, we split the feedback process into two screens (Figure 3b & c) and note that this obviates the player confusion seen in our initial study.

**Updating the interface** The second observational study highlighted three major concerns. First, the actual *Disco Elysium* game provides visual cues that were missing from the web interface that players relied upon to follow the conversation. We address this concern by tweaking our interface to better match the one from the game, including highlighting actor names using the same colors from the game and updating the icons for the player statistics shown at the bottom of the screen in Figure 3a. Second, we decided to instruct players to exhaust all dialogue options within the conversation, which allows us to improve our coverage of the generated utterances in the dialogue graph. This process differs from the actual gameplay, in which players are free to skip through many dialogues that serve to provide a backstory to the game world. Finally, it became clear that players might not complete the annotation task within one session, so we added support for automatically saving and resuming the

task starting where the player left off.

## F Predefined Tags

We manually code the free-form comments from our first pilot study to understand common justifications for expressed preferences. We do this both for comments on a player's initial preference and upon retroactively updating their preference based upon seeing the next utterance. We then update our interface to include our manually coded categorizations as predefined tags that players can select to justify their choices. For the full list of predefined tags and the description we provide for the tag in our interface, please see Table A6 (initial preference) and Table A7 (upon retrospectively changing preference).

| Predefined Tag | Description |
|---|---|
| `better advances my goals` | After seeing the next line of dialogue, it turns out that the other dialogue option better advances your goals. |
| `better explores my options` | After seeing the next line of dialogue, it turns out that the other dialogue option better explores the conversation. |
| `better matches desired mood` | After seeing the next line of dialogue, it turns out that the other dialogue option better matches the mood your are going for. |
| `contains more specifics` | After seeing the next line of dialogue, it turns out that the other dialogue option actually contains more context specific information. |
| `illogical in hindsight` | After seeing the next line of dialogue, it turns out that the other dialogue option does not make sense in context. |
| `is a repeat in hindsight` | After seeing the next line of dialogue, it turns out that the other dialogue option repeats something that was previously stated. |
| `seems more interesting now` | After seeing the next line of dialogue, it turns out that the other dialogue option is actually more interesting. |
| `other` | Please explain in your own words why you changed the selected dialogue option. |

Table A7: List of predefined tags and their associated description from our web app for justifying why a player *retrospectively changed their preference*.

# G Rating Trends

Due to the nature of the game using a dialogue graph where node reachability is altered depending on the dialogue option chosen, on average each player only rates 41 utterances out of the 112 unique utterances rated by all players (Table 4). To understand how the player preference changes as the number of ratings a particular utterance receives, we produce stacked histograms (Figure A1 & Figure A2) where each bar represents player preference given the number of players rating an utterance.

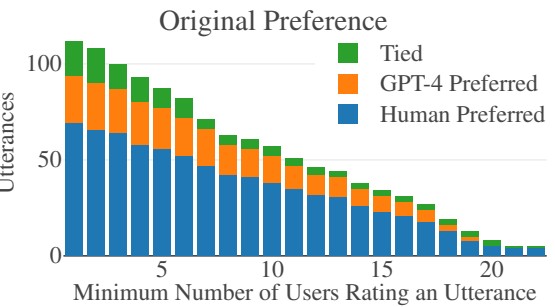

Figure A1: Histogram of *initial candidate* preferences based on minimum number of players rating the utterance.

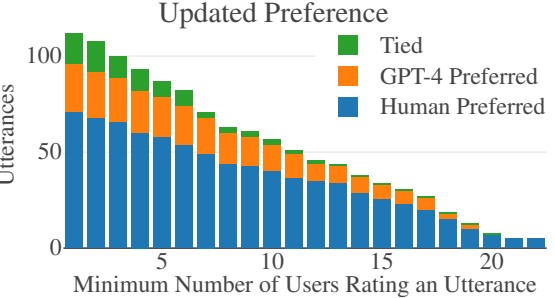

Figure A2: Histogram of *retroactively updated* preferences based on minimum number of players rating the utterance.