# OpenReview forum: "A Framework for Exploring Player Perceptions of LLM-Generated Dialogue in Commercial Video Games"
_EMNLP/2023/Conference — EMNLP 2023 Findings_

### Official Review · Reviewer_NDsa · 2023-07-31

**Soundness:** 4

**Excitement:**

3: Ambivalent: It has merits (e.g., it reports state-of-the-art results, the idea is nice), but there are key weaknesses (e.g., it describes incremental work), and it can significantly benefit from another round of revision. However, I won't object to accepting it if my co-reviewers champion it.

**Missing References:**

Suggestions for references for related work about comparing LLM output with human output and papers that have studied what issues there are with LLM-generated dialog contributions:

Belinda Z Li, Maxwell Nye, and Jacob Andreas. 2022. Language modeling with latent situations. arXiv preprint arXiv:2212.10012.

Jacob Andreas. 2022. Language models as agent models. In Findings of the Association for Computational Linguistics: EMNLP 2022, pages 5769–5779, Abu Dhabi, United Arab Emirates. Association for Computational Linguistics.

Yusuke Mori, Hiroaki Yamane, Yusuke Mukuta, and Tatsuya Harada. 2019. Toward a better story end: Collecting human evaluation with reasons. In Proceedings of the 12th International Conference on Natural Language Generation, pages 383–390, Tokyo, Japan. Association for Computational Linguistics.

**Paper Topic And Main Contributions:**

This paper describes a user study in which LLM output is compared to human expert-generated text in a specific game context (the Disco Elysium game). The purpose is to get insight into LLMs' application to the video game domain. The games that are targetted are role-playing video games in which different characters produce contributions depending on the game state and the character state.

The evaluation of LLM output is done by letting a small group of players play a specific video game and choosing their preferred dialog contribution in certain situations, as well as asking for their reasons for the choice (both with pre-defined and freetext reasons).

The main contributions are:

- A task design for comparing GPT-4 generated output to human expert output for the Disco Elysium game.

- A user study that includes a specially designed interface, carefully designed rating options verified via an observation study and a specific target group of raters that were recruited to spend a considerable amount of time in order to gather data for analysis.

- The analysis of the gathered data, including some insights into why raters chose one option or the other.

**Questions For The Authors:**

A: How were raters instructed and did they know what the purpose of their ratings was and even that some contributions were machine-generated? Were they asked about their opinion of machine-generated content in general?

B: What is the role of the Lua scripts? Is there any particular reason that it is Lua scripts rather than some other programming language or just natural language? Is there any reason to believe that Lua scripts are better suited as input for an LLM than another formalization?

C: Are there any demographics (e.g. age, gender) about the participants in the usability studies that you can report? Do the demographics match what is known about the demographics of people actually playing the game?

D: For the contributions for which you have 3 or more ratings, how much do raters agree in their decisions? Even if 100 instances is small, this can give a first glimpse into how generalizable the ratings are for the game and this target group.

**Reasons To Accept:**

- The task setup is interesting and meaningful in the application evaluation that is targetted here. The main question that the authors are trying to answer is whether LLMs can be used for generated character contributions in video games and the setup tests exactly that by using the actual game variables to elicit GPT-4 output.

- The work compares GPT-4 output with human expert output in a well-defined context that stays as close to the final target domain as possible: a commercial video game.

- The study is extensive in that it required recruiting a specific target group of participants that then needed to spend substantial time playing and rating contributions in order to gather a meaningful amount of data.

**Reasons To Reject:**

- The evaluation setup is rather specific to the game that is described here and it is unclear whether and how this can be tested with other games. Even though the limitations section mentions the limited generalizability, I consider this a serious issue for publication.

- Even though substantial effort is put into data collection, the final dataset that is analyzed is small, only a hundred contributions receive 3 judgments, making the results rather weak.

- In addition, there are factors that make the already small rater group heterogenous, such as play style, experience with the game, and possibly demographic factors (that are impossible to know here). This adds to the low generalizability.

- Since only raters who have already played the game were included in the study (with good reason), it is unclear to me what role familiarity bias would play when rating dialog contributions. Being used to a certain story line might make raters prefer contributions they have seen before or chosen before.

- The related works section mentions only work related to video games. I would like to see here some related work about comparing LLM output with human output and references to papers that have studied what issues there are with LLM-generated dialog contributions.

*Update after rebuttal*

Thank you to the authors for clearly addressing my concerns. I would like to see the details on power analysis as well as additional background references in a final version of the paper.

**Reproducibility:**

1: Could not reproduce the results here no matter how hard they tried.

**Reviewer Confidence:**

4: Quite sure. I tried to check the important points carefully. It's unlikely, though conceivable, that I missed something that should affect my ratings.

**Typos Grammar Style And Presentation Improvements:**

- Figure 1: The text examples are too small, please increase as much as possible with the limited space.

- p.5, bottom: "partcipant" -> "participant"

- Figure 3: I do not understand whether these graphs show all ratings or just a subset (cf. Sec 5.1, first paragraph)

---

> ### Author Rebuttal · Authors · 2023-08-26
>
> Under reasons to accept, you describe our work as interesting and meaningful.
> We're thankful you noticed!
> We appreciate your feedback and sincerely wish that you'll read our comments and reflect on them during the reviewer discussion period.
> If you do, we hope that you will update your score and associated review text to indicate how our comments below factor into your decisions.
>
> **Background**
>
> We wish to bring your attention to an important consideration for our work: while many companies are looking to integrate LLMs into video games, no published research has performed an extrinsic evaluation of model-generated dialogue in a video game, not even from major game studios like Microsoft [[1](https://arxiv.org/abs/2212.10618)] who have direct access to the game code.
> The referenced paper (which we only became aware of after submission and will cite in a revision of the paper) explores dialogue generation for the roleplaying video game *The Outer Worlds*, but in contrast only looks at utterances **without conditioning on game state** and uses a mix of automatic metrics and **out-of-context human evaluation**.
> In comparison, we reverse engineer the dialogue system from *Disco Elysium* and recreate it in a web app environment which reduces confounding variables (e.g. choosing where to go, who to interact with, etc).
> **The release of our code and tools will democratize the ability to conduct future in-context human evaluation for research exploring interactive dialogue in video games.**
> This is important since LLMs are actively being investigated for video games and other virtual experiences as mentioned in our paper: many well-funded startups like InWorld AI [[2](https://inworld.ai/blog/inworld-valued-at-500-million)], character.ai [[3](https://www.crunchbase.com/organization/character-ai)], and convai [[4](https://investor.nvidia.com/news/press-release-details/2023/NVIDIA-ACE-for-Games-Sparks-Life-Into-Virtual-CharactersWith-Generative-AI/default.aspx)] have begun offering products in this space, and established gaming companies like Microsoft [[1](https://arxiv.org/abs/2212.10618)] and Ubisoft [[5](https://aclanthology.org/2023.nlp4convai-1.11/)] are conducting research in the area too.
>
> **Generalizability**
>
> While we do mention that our work does not generalize to all narrative-driven games, the framework we put forth is indeed useful across many games: any video game that provides dialogue choices conditioned on game state.
> These types of games are played by millions of players: see Baldur's Gate 3 [[6](https://wccftech.com/baldurs-gate-3-topped-5-2-million-units-sold-on-steam-says-belgian-embassy/)] as a very recent example of their popularity (see also the examples we highlight in our **Background** above).
> Note that *Disco Elysium* uses an off-the-shelf dialogue system made for the Unity game engine which is in use by many additional games [[7](https://www.pixelcrushers.com/games-showcase/), [8](https://www.articy.com/en/showcases/)].
> This means the tools we release can be directly used to to explore and conduct experiments on a wide range of currently released video games.
>
> **Study Size and Heterogeneous Raters**
>
> We appreciate the concern over the study size in the face of potentially heterogeneous raters.
> As you note, our paper describes a study where 100 utterances receive at least three annotations across our 28 participants as discussed in our **Study parameters** in Section 4.3.
> Using the methodology from [[9](https://aclanthology.org/2020.emnlp-main.745/)], our study design has a statistical power of 0.96 for a margin of 10% in a high variance population (see the tables at the end for a more complete analysis).
> The observed margins in our study as described in Section 5.1 are often much larger (e.g. in aggregate 61% of players prefer human-written dialogue, a margin of 22%) indicating more than sufficient statistical power for our reported results (a power $\geq0.8$ is typically desired).
> The additional fine grained analysis in Section 5.2 covers specific actionable areas of concern to improve modeling for the task in the future.
> Therefore we believe the scale of the study is adequate for the findings we report even across differences such as play style.
> We will include a discussion of our study's statistical power in a revision of the paper.
>
> **Familiarity Bias**
>
> As described in Section 4.3, players need to be experienced with the game because:
>
> * Dialogue choices in *Disco Elyisum* affect the game state and change what dialogue options the player has available in the future.
> This is an acquired skill we ensure participants have by limiting participation to those who have played the game.
> * The conversation players are evaluating are from our validation set (the dataset split was automatically determined by minimizing game state overlap as described in Section 2.1).
> This conversation is from late in the game and thus requires familiarity with the game to understand and assess the conversation, e.g. we wouldn't expect someone to understand references in the last episode of a TV show if they haven't watched the rest of the show; the same consideration applies here.
>
> Please note that from our pilot and observational study it became clear that players could not remember exact dialogue options.
> As the game was originally released in 2019 with an updated release in 2021, most players had completed their play-through significantly earlier than our main study's timeframe in Spring of 2023.
>
> **Additional References**
>
> We would be happy to include a discussion of additional references and how they relate to our work.
>
> **Question A: Additional Instructions**
>
> In addition to the instructions seen in Figure 2 of our paper, raters were given the following instructions:
>
> > As you play, you will be presented with dialogue options that may be machine-generated, paired alongside the original dialogue from the game (e.g. *1a* vs *1b*).
> > Anytime you choose a paired dialogue option you will be prompted to state why you chose the option.
> > You will then be presented the next line of dialogue that will be spoken and asked to reassess your choice.
> > Please consider the next line of dialogue and the historical context of the dialogue thus far when justifying your choices (e.g., because your choice fit the context better than the other choice or was more interesting).
> > * Please play through the scenario as many times as needed to exhaust all the dialogue options.
> > * We highly encourage you to include quotations from the dialogue history or the choices in your justifications!
>
> **Question B: Why Lua?**
>
> We use Lua to ground the dialogue with game state as the *Disco Elysium* dialogue system is Lua-based and there is no way to directly translate Lua code into natural language.
> Therefore we focus on models which have shown strong performance on both code and language domains like GPT-4.
> Additionally, by formalizing the dialogue structure as Lua we open the possibility to conduct future research that allows for dynamically changing the game state based on machine-generated utterances, an open problem.
>
> **Question C: Demographics**
>
> We specifically do not collect demographic information in order to conform to the requirements of our Institutional Review Board (IRB), only ensuring that participants are at least 18 years of age and have completed a play-through of the game.
>
> **Question D: Study Generalizability**
>
> Please refer to our discussion of **Study Size and Heterogeneous Raters** above.
>
> **Thanks for reading!**
>
> In light of our comments addressing your concerns, we sincerely hope you'll consider raising both your soundness and excitement scores.
> Specifically, we believe **we provide sufficient support for all of our claims** and that **our work will enable a promising new research direction** for evaluating video game dialogue in an interactive setting, which is an area of growing interest [[1](https://arxiv.org/abs/2212.10618),[2](https://inworld.ai/blog/inworld-valued-at-500-million),[3](https://www.crunchbase.com/organization/character-ai),[4](https://investor.nvidia.com/news/press-release-details/2023/NVIDIA-ACE-for-Games-Sparks-Life-Into-Virtual-CharactersWith-Generative-AI/default.aspx),[5](https://aclanthology.org/2023.nlp4convai-1.11/)].
>
>
> \[1\]: https://arxiv.org/abs/2212.10618
> \[2\]: https://inworld.ai/blog/inworld-valued-at-500-million
> \[3\]: https://www.crunchbase.com/organization/character-ai
> \[4\]: https://investor.nvidia.com/news/press-release-details/2023/NVIDIA-ACE-for-Games-Sparks-Life-Into-Virtual-CharactersWith-Generative-AI/default.aspx
> \[5\]: https://aclanthology.org/2023.nlp4convai-1.11/
> \[6\]: https://wccftech.com/baldurs-gate-3-topped-5-2-million-units-sold-on-steam-says-belgian-embassy/
> \[7\]: https://www.pixelcrushers.com/games-showcase/
> \[8\]: https://www.articy.com/en/showcases/
> \[9\]: https://aclanthology.org/2020.emnlp-main.745/
>
> We report the results of the statistical power analysis of our study design below:
> ```
> Num Workers: 10
> Margin | Power | Variance
> 0.05   | 0.765 | low
> 0.05   | 0.325 | high
> 0.1    | 1     | low
> 0.1    | 0.775 | high
> 0.2    | 1     | low
> 0.2    | 1     | high
> ```
>
> ```
> Num Workers: 28
> Margin | Power | Variance
> 0.05   | 0.91  | low
> 0.05   | 0.475 | high
> 0.1    | 1     | low
> 0.1    | 0.96  | high
> 0.2    | 1     | low
> 0.2    | 1     | high
> ```
>
> ```
> Num Workers: 50
> Margin | Power | Variance
> 0.05   | 0.925 | low
> 0.05   | 0.61  | high
> 0.1    | 1     | low
> 0.1    | 1     | high
> 0.2    | 1     | low
> 0.2    | 1     | high
> ```

---

### Official Review · Reviewer_aXdf · 2023-08-04

**Soundness:** 3

**Ethical Concerns:**

Yes

**Excitement:**

4: Strong: This paper deepens the understanding of some phenomenon or lowers the barriers to an existing research direction.

**Justification For Ethical Concerns:**

Not a major red flag, and the authors do discuss it in the appropriate section, but I still wanted to note here that I am not fully satisfied with that discussion. The underlying material (the dialogue tree from the game) is copyrighted material, and the creators of that material did not consent to it being used for this study. This is the case for much of the material that we work with, but when the material comes from a single author or a small set of authors, it is thrown in even sharper relief.

**Paper Topic And Main Contributions:**

Paper presents a new resource (the extracted dialogue tree from a video game) and a new task ("dialogue infilling", generating a game dialogue utterance given the game state), and evaluates GPT-4 on this task (for human preference), showing that the original utterance is mostly preferred.

- NLP engineering experiment
- new data resource

**Questions For The Authors:**

A: Can you please state again what you think is the generalisable take-home message from your experiment? What have we learned about the capabilities of LLMs?

B: Please explain again why the evaluators had to be familiar with the game.

C: Section 4.3, how much was payed for the study alltogether? How many gift cards were offered?

**Reasons To Accept:**

- interesting resource (human-authored dialogue game tree + game state extracted from game) + human preference ranking over original and LLM generated dialogue utterances in context

- interesting task and experiment setup ("dialogue infilling", generate utterance that fits to given state, compare with original utterance from that point)

**Reasons To Reject:**

- relies on intuitive notion that the task is hard and somehow involves language understanding, but doesn't spend any time analysing data for linguistic features of human-authored text vs computer generated one. What exactly is it that makes this task hard? Why would we even think to begin with that LLMs could handle the state description?

- relatively little effort is spent on setting up the model for tackling the task, and very much effort on evaluating it. While in general I appreciate that, this limits the conclusions that can be drawn. If the focus was on creating a model that can do this task, I would have expected to see ablation studies, or more variation on how the game state is communicated to the model (e.g., not as LUA code but in natural language).

- the evaluators were familiar with the game. Could it not be that the performed style identification rather than really were judging appropriateness?

- the experiments are not freely replicable, as the data source is copyrighted material.

**Reproducibility:**

3: Could reproduce the results with some difficulty. The settings of parameters are underspecified or subjectively determined; the training/evaluation data are not widely available.

**Reviewer Confidence:**

4: Quite sure. I tried to check the important points carefully. It's unlikely, though conceivable, that I missed something that should affect my ratings.

**Typos Grammar Style And Presentation Improvements:**

- not sure whether word-count(Atlas Shrugged) should count as unit for measuring length of written material

- line 386, and elsewhere: you talk about "LLM-generated dialogue", but it seems to me that you at most mean "LLM-generated dialogue utterances"

---

> ### Author Rebuttal · Authors · 2023-08-26
>
> Under reasons to accept, you state that our dataset, task, and experiment design are interesting.
> Thanks for the kind words!
> We appreciate your feedback and sincerely wish that you'll read our comments and reflect on them during the reviewer discussion period.
> If you do, we hope that you will update your score and associated review text to indicate how our comments below factor into your decisions.
>
> **Background**
>
> We wish to bring your attention to an important consideration for our work: while many companies are looking to integrate LLMs into video games, no published research has performed an extrinsic evaluation of model-generated dialogue in a video game, not even from major game studios like Microsoft [[1](https://arxiv.org/abs/2212.10618)] who have direct access to the game code.
> The referenced paper (which we only became aware of after submission and will cite in a revision of the paper) explores dialogue generation for the roleplaying video game *The Outer Worlds*, but in contrast only looks at utterances **without conditioning on game state** and uses a mix of automatic metrics and **out-of-context human evaluation**.
> In comparison, we reverse engineer the dialogue system from *Disco Elysium* and recreate it in a web app environment which reduces confounding variables (e.g. choosing where to go, who to interact with, etc).
> **The release of our code and tools will democratize the ability to conduct future in-context human evaluation for research exploring interactive dialogue in video games.**
> This is important since LLMs are actively being investigated for video games and other virtual experiences as mentioned in our paper: many well-funded startups like InWorld AI [[2](https://inworld.ai/blog/inworld-valued-at-500-million)], character.ai [[3](https://www.crunchbase.com/organization/character-ai)], and convai [[4](https://investor.nvidia.com/news/press-release-details/2023/NVIDIA-ACE-for-Games-Sparks-Life-Into-Virtual-CharactersWith-Generative-AI/default.aspx)] have begun offering products in this space, and established gaming companies like Microsoft [[1](https://arxiv.org/abs/2212.10618)] and Ubisoft [[5](https://aclanthology.org/2023.nlp4convai-1.11/)] are conducting research in the area too.
>
> **Modeling Concerns**
>
> We want to first address your concern regarding our focus on evaluation over modeling for this task.
> We greatly appreciate that you notice the considerable effort we spent on our evaluation!
> We believe this aligns very closely with the theme of the *Human-Centered* NLP track our paper was submitted to.
> When this track was first announced as a special track at NAACL 2022 [[6](https://2022.naacl.org/blog/special-theme/)], it was made clear that human factors are the focus rather than modeling contributions (emphasis added):
> > If the submission contributes a new modeling or training technique that is evaluated on standard corpus-based benchmarks, it would best fit the regular track. **However, if the submission evaluates, e.g., how users calibrate their trust in the predictions of the system, it would be a good fit for the theme track.**
>
> Our player-centered analysis in Section 5 clearly aligns with the stated goal above.
>
> That said, while we do not focus heavily on the modeling for this task, we experiment with both fine-tuning and few-shot approaches as described in Appendix C.
> We use Lua to ground the dialogue with game state as the *Disco Elysium* dialogue system is Lua-based and there is no way to directly translate Lua code into natural language.
> Therefore we focus on models which have shown strong performance on both code and language domains like GPT-4.
> Additionally, by formalizing the dialogue structure as Lua we open the possibility to conduct future research that allows for dynamically changing the game state based on machine-generated utterances, an open problem.
>
> **Linguistic Analysis**
>
> As you can see from Section 5.2, we include a fine-grained analysis performed by players comparing human-written and machine-generated dialogue.
> This analysis includes a discussion of observed linguistic features, such as awkward articulations (e.g. combining the use of first person and third person pronouns within a single utterance) and interjecting spurious phrases during paraphrasing.
>
> **Style Identification**
>
> If you look at the results highlighted in Figure 3b and discussed in Section 5.1, players are not merely performing style identification.
> The most common reason for changing their preference from LLM-generated to human-written dialogue is that the utterance is illogical in context.
> Though since our paper is concerned with *human-centered* factors, even if players do perform style identification, that would still be an important consideration: one of the largest video game publishers, Ubisoft, is actively researching style for video game dialogue to ensure consistency [5].
>
> **Ethics**
>
> In regards to your ethical concerns, we take great care to respect the copyright of the work.
> We want to make it clear: **we do not directly release a dataset**.
> Rather, we release instructions for using AssetStudio [[7](https://github.com/Perfare/AssetStudio)] to extract the game data from a **legally obtained copy of the game**.
> We then provide tools for processing this data allowing easier replication and enabling future experimentation by researchers.
> Finally, our study requires a login with a registered account only available to participants who have previously completed the game.
>
> **Question A: Take-home Message**
>
> As we highlighted in the **Background** above, LLMs are actively being investigated for video games.
> Despite this interest, there is no published research on player experience interacting with the LLMs in a game setting.
> Our research provides both macro- and fine-grained analysis of specific issues LLMs face in this context.
>
> **Question B: Familiarity**
>
> As described in Section 4.3, players need to be experienced with the game because:
>
> * Dialogue choices in *Disco Elyisum* affect the game state and change what dialogue options the player has available in the future.
> This is an acquired skill we ensure participants have by limiting participation to those who have played the game.
> * The conversation players are evaluating are from our validation set (the dataset split was automatically determined by minimizing game state overlap as described in Section 2.1).
> This conversation is from late in the game and thus requires familiarity with the game to understand and assess the conversation, e.g. we wouldn't expect someone to understand references in the last episode of a TV show if they haven't watched the rest of the show; the same consideration applies here.
>
> Please note that from our pilot and observational study it became clear that players could not remember exact dialogue options.
> As the game was originally released in 2019 with an updated release in 2021, most players had completed their play-through significantly earlier than our main study's timeframe in Spring of 2023.
>
> **Question C: Study Cost**
>
> We had 5 participants for our pilot study, 2 for our observational study, and 28 for our main study.
> In total, we spent nearly 1000 dollars in gift cards across all participants (a breakdown of gift card amounts is detailed in Section 9).
> We further spent approximately 3000 dollars in cloud costs for both our fine-tuning and few-shot experiments.
> Finally, we spent 1.5 years of research time to produce this study, starting from Jan 2022 -- an exact salary breakdown is difficult to provide.
>
> **Thanks for reading!**
>
> In light of our comments addressing your concerns, we sincerely hope you'll consider raising both your soundness and excitement scores.
> Specifically, we believe **we provide sufficient support for all of our claims** and that **our work will enable a promising new research direction** for evaluating video game dialogue in an interactive setting, which is an area of growing interest [[1](https://arxiv.org/abs/2212.10618),[2](https://inworld.ai/blog/inworld-valued-at-500-million),[3](https://www.crunchbase.com/organization/character-ai),[4](https://investor.nvidia.com/news/press-release-details/2023/NVIDIA-ACE-for-Games-Sparks-Life-Into-Virtual-CharactersWith-Generative-AI/default.aspx),[5](https://aclanthology.org/2023.nlp4convai-1.11/)].
>
>
> \[1\]: https://arxiv.org/abs/2212.10618
> \[2\]: https://inworld.ai/blog/inworld-valued-at-500-million
> \[3\]: https://www.crunchbase.com/organization/character-ai
> \[4\]: https://investor.nvidia.com/news/press-release-details/2023/NVIDIA-ACE-for-Games-Sparks-Life-Into-Virtual-CharactersWith-Generative-AI/default.aspx
> \[5\]: https://aclanthology.org/2023.nlp4convai-1.11/
> \[6\]: https://2022.naacl.org/blog/special-theme/
> \[7\]: https://github.com/Perfare/AssetStudio

---

### Official Review · Reviewer_h39F · 2023-08-04

**Soundness:** 4

**Excitement:**

4: Strong: This paper deepens the understanding of some phenomenon or lowers the barriers to an existing research direction.

**Missing References:**

- 043: Footnotes/references can be inserted for "AI Dungeon, InWorld AI, and ConvAI".
- 122: Footnote with link (https://www.lua.org/) or reference can be added for "Lua" language scripts/expressions used throughout the text for non-familiar readers.
- 136: Footnote/reference can be inserted for "Atlas Shrugged".

**Paper Topic And Main Contributions:**

- This paper focuses on evaluating the impacts of LLM-generated dialogue on the player experience.
- The authors asked the gamers to interact with LLM-generated dialogue injected into 'Disco Elysium: The Final Cut', a widely-popular dialogue-centered role-playing game.
- This study investigates a constrained dialogue generation task in which an LLM must decide how to update a dialogue to match the corresponding game state.
- For this dialogue infilling task, an LLM receives human-written dialogue text as input and tweaks it to fit various dynamic aspects of the game state (i.e., GPT-4 generates masked utterances conditioned upon the game state).
- The authors evaluated GPT-4 LLM outputs via an acceptable-size user study, asking 28 Disco Elysium players to provide preference judgments and free-form feedback using a web interface to mimic the game's conversation functionality.
- The findings suggest that players strongly prefer the original game engine dialogue over the LLM-generated dialogue due to better logical consistency and flow with the game state.
- Participants also note that GPT-4 can close the gap by providing creative and interesting dialogue options. However, it often does that at the expense of logical flow and game-state grounding.
- To contribute to future research in player-centered video game dialogue generation, the authors release the web-based annotation interface and tools to reproduce similar game datasets.

--- Updates After Rebuttal ---
- Reworded "a large-scale user study" with "an acceptable-size user study"(appreciate the authors's response: "We will include a discussion of our study's statistical power in a revision of the paper and will happily address your concern by not referring to our study as "large-scale".")

**Questions For The Authors:**

- Table 1 (content & caption): What is a dialogue fork? Please define such specific terms briefly for non-familiar readers.
- 100-104 & 409-417 & 427-429: What do "H:" vs. "G:" stand for in these paragraphs? Please clarify (e.g., "H" for human-written dialogue vs. "G" for GPT-4 generations).
- How deep do you go in conversation history for the GPT-4 generated responses? Do you provide the whole game session dialogue history, or is there a fixed number of conversational turns to go back and check (for efficiency)? Please clarify.

--- Updates After Rebuttal ---
- Appreciate the clarifications for "Terminology" and "Undefined initials".
- Regarding the "Dialogue History", I was referring to your observations in lines 441-449 (see below) and wondering if including the dialogue history (maybe up to 10 turns for efficiency) would help to resolve these issues:
"Inconsistencies with the game world: Many of the justifications for players’ preferences mention appropriateness or logical consistency / flow with the conversation history. In general, the world created by the game designers has a depth and consistency that is difficult for models to understand, especially when it is encoded programmatically (e.g., in Lua scripts) rather than in unstructured text."

**Reasons To Accept:**

- This study performs an acceptable user study of LLM-generated dialogue augmented into video games, with 28 players of 'Disco Elysium: The Final Cut' providing valuable insights on player experience.
- The authors mainly examine the following critical research question: Can an LLM understand enough about the game state to appropriately modify dialogue in a way that is logically and tonally consistent with the game state?
- They focus on a constrained dialogue generation task, in which the game state is integrated into the GPT-4 LLM few-shot prompt via Lua scripts that encode game variables, functions, and dialogue.
- This constrained setup makes the task more tractable for evaluation and practically relevant to future human-AI/LLM collaboration in interactive game/storytelling applications.
- The authors implemented a web interface to recreate Disco Elysium's dialogue engine to perform these evaluations, and this interface and tools are released to enable researchers to build upon this initial work.
- Results indicate that human-written dialogue is preferred strongly over GPT-4 LLM generations for reasons like improved logical flow, appropriateness, and tonal consistency.
- Overall, the paper is well-written and clearly structured for readability.

--- Updates After Rebuttal ---
- Reworded "a large-scale user study" with "an acceptable user study" (appreciate the authors's response regarding the "Study Size").

**Reasons To Reject:**

- The authors suggest that large number of players required for adequate coverage of the game's dialogue graph. They also claim a large-scale user study is necessary to evaluate the LLM-generated responses in this setup. While recruiting 28 players is not a small number, and it requires a great effort to build a system like this to collect such valuable data, due to the highly subjective nature of players experiences, one can argue that calling it a "large-scale" user study may not be well-accepted in this context. While there are no hard and fast rules around how many people you should involve in this type of research, having a population size of hundreds (even thousands) would be more appropriate to call it a "large-scale" user study comfortably.

- The LLM-generated dialogue text shown to the participants are pre-generated from GPT-4, so each of the 28 players rates the same generated utterances. That indicates the lack of considering player's personal choices and specific paths on the game dialogue graph during the evaluation process. In the future, it would be better to explore methods to dynamically generate these utterances and eventually reflect the interactive nature of such video games.

- No significant flaws detected for rejection, but upon acceptance, the authors must ensure that the web interface and tools to crawl/annotate similar datasets will be publicly available.

--- Updates After Rebuttal ---
- The discussions around "Study Size" was helpful and convincing to some degree (in the revised version, adding more discussions on statistical significance of your results based an the adequacy of population size would definitely help)
- Regarding the "Dynamic Text Generation", I appreciate the reasoning behind this choice (as partly discussed in the Limitations section). However, it would help to emphasize this more in the main paper and discuss why this is important (but difficult and costly in terms of variance and population size) for a more realistic setup and evaluation in the future (could be part of the Conclusion/Future Work, not just the Limitations).

**Reproducibility:**

2: Would be hard pressed to reproduce the results. The contribution depends on data that are simply not available outside the author's institution or consortium; not enough details are provided.

**Reviewer Confidence:**

3: Pretty sure, but there's a chance I missed something. Although I have a good feel for this area in general, I did not carefully check the paper's details, e.g., the math, experimental design, or novelty.

**Typos Grammar Style And Presentation Improvements:**

Minor issues:
- 045, 052, 133, 180, 182, 218, 256, 288, 331, 342, 351, 360, 635: ".^n" could be "^n." (placing the footnote numbers - in superscripts - before the punctuation).
- 351 (footnote 13): "IRB" could be "Institutional Review Board (IRB)".
- Section titles: Inconsistent capitalization exists in main section titles (please check the template or choose one style for consistency).
- Tables 1, 2, 3, 4: Text font sizes used in table content could be smaller (for improved presentation style and consistency, plus potentially to gain some space for further discussions).
- 568-569: "to the best of our knowledge no commercial video games" should be "to the best of our knowledge, no commercial video games" (missing comma).
- 790: "Disco Elysiumdataset" should be "Disco Elysium dataset" (missing space).

---

> ### Author Rebuttal · Authors · 2023-08-26
>
> Under reasons to accept, you describe our work as providing valuable insights.
> Thanks for noticing!
> We appreciate your feedback and sincerely wish that you'll read our comments and reflect on them during the reviewer discussion period.
> If you do, we hope that you will update your score and associated review text to indicate how our comments below factor into your decisions.
> Thanks!
>
> **Background**
>
> We wish to bring your attention to an important consideration for our work: while many companies are looking to integrate LLMs into video games, no published research has performed an extrinsic evaluation of model-generated dialogue in a video game, not even from major game studios like Microsoft [[1](https://arxiv.org/abs/2212.10618)] who have direct access to the game code.
> The referenced paper (which we only became aware of after submission and will cite in a revision of the paper) explores dialogue generation for the roleplaying video game *The Outer Worlds*, but in contrast only looks at utterances **without conditioning on game state** and uses a mix of automatic metrics and **out-of-context human evaluation**.
> In comparison, we reverse engineer the dialogue system from *Disco Elysium* and recreate it in a web app environment which reduces confounding variables (e.g. choosing where to go, who to interact with, etc).
> **The release of our code and tools will democratize the ability to conduct future in-context human evaluation for research exploring interactive dialogue in video games.**
> This is important since LLMs are actively being investigated for video games and other virtual experiences as mentioned in our paper: many well-funded startups like InWorld AI [[2](https://inworld.ai/blog/inworld-valued-at-500-million)], character.ai [[3](https://www.crunchbase.com/organization/character-ai)], and convai [[4](https://investor.nvidia.com/news/press-release-details/2023/NVIDIA-ACE-for-Games-Sparks-Life-Into-Virtual-CharactersWith-Generative-AI/default.aspx)] have begun offering products in this space, and established gaming companies like Microsoft [[1](https://arxiv.org/abs/2212.10618)] and Ubisoft [[5](https://aclanthology.org/2023.nlp4convai-1.11/)] are conducting research in the area too.
>
> **Study Size**
>
> Your first stated concern is whether our usage of the term "large-scale" is appropriate.
> Underlying that concern is an important question: Are 28 players enough to provide meaningful insights into our research question regarding player experience with LLM-generated dialogue?
> We begin by noting that our paper ensures that 100 utterances receive at least three annotations across our 28 participants as discussed in our **Study parameters** in Section 4.3.
> Using the methodology from [[6](https://aclanthology.org/2020.emnlp-main.745/)], our study design has a statistical power of 0.96 for a margin of 10% in a high variance population (see the tables at the end for a more complete analysis).
> The observed margins in our study as described in Section 5.1 are often much larger (e.g. in aggregate 61% of players prefer human-written dialogue, a margin of 22%) indicating more than sufficient statistical power for our reported results (a power $\geq0.8$ is typically desired).
> The additional fine grained analysis in Section 5.2 covers specific actionable areas of concern to improve modeling for the task in the future.
> Therefore we believe the scale of the study is adequate for the findings we report.
> We will include a discussion of our study's statistical power in a revision of the paper and will happily address your concern by not referring to our study as "large-scale".
>
> **Dynamic Text Generation**
>
> Your second concern is whether our evaluation is dynamic enough.
> We discuss why we make this choice in our limitations section.
> Including a player's dialogue history as part of the prompt is an easy change to include in our framework, but we chose not to do so as it may introduce a large amount of variance necessitating a considerably larger number of participants to achieve a similar statistical power as our current design.
> This is certainly a task that can be explored in future research once our code is released.
> Though please note that our work goes above and beyond any other research in the field as discussed above [[1](https://arxiv.org/abs/2212.10618)].
>
> **Question 1: Terminology**
>
> A dialogue fork is a node with an outgoing edge where traversal is gated by a binary game state expression -- the edge can only be traversed if the game state matches a given Lua condition.
> We will clarify such terms further in a revision of the paper.
>
> **Question 2: Undefined Initials**
>
> You are correct, *H* is for human and *G* is for GPT-4 in those paragraphs.
> We will clarify this in a revision of the paper.
>
> **Question 3: Dialogue History**
>
> We are not sure we fully understand the question, but will try our best to answer.
> The prompts we use for dialogue generation do not contain any dialogue history, rather they include the game writer's comments and the alternate dialogue options conditioned upon game state.
> For an example, please see the Lua script in Step 2 of Figure 1.
> In regards to player annotation, we allow players to edit their feedback for a particular utterance after seeing future utterances.
>
> **Thanks for reading!**
>
> In light of our comments addressing your concerns, we sincerely hope you'll consider raising both your soundness and excitement scores.
> Specifically, we believe **we provide sufficient support for all of our claims** and that **our work will enable a promising new research direction** for evaluating video game dialogue in an interactive setting, which is an area of growing interest [[1](https://arxiv.org/abs/2212.10618),[2](https://inworld.ai/blog/inworld-valued-at-500-million),[3](https://www.crunchbase.com/organization/character-ai),[4](https://investor.nvidia.com/news/press-release-details/2023/NVIDIA-ACE-for-Games-Sparks-Life-Into-Virtual-CharactersWith-Generative-AI/default.aspx),[5](https://aclanthology.org/2023.nlp4convai-1.11/)].
>
>
> \[1\]: https://arxiv.org/abs/2212.10618
> \[2\]: https://inworld.ai/blog/inworld-valued-at-500-million
> \[3\]: https://www.crunchbase.com/organization/character-ai
> \[4\]: https://investor.nvidia.com/news/press-release-details/2023/NVIDIA-ACE-for-Games-Sparks-Life-Into-Virtual-CharactersWith-Generative-AI/default.aspx
> \[5\]: https://aclanthology.org/2023.nlp4convai-1.11/
> \[6\]: https://aclanthology.org/2020.emnlp-main.745/
>
> We report the results of the statistical power analysis of our study design below:
> ```
> Num Workers: 10
> Margin | Power | Variance
> 0.05   | 0.765 | low
> 0.05   | 0.325 | high
> 0.1    | 1     | low
> 0.1    | 0.775 | high
> 0.2    | 1     | low
> 0.2    | 1     | high
> ```
>
> ```
> Num Workers: 28
> Margin | Power | Variance
> 0.05   | 0.91  | low
> 0.05   | 0.475 | high
> 0.1    | 1     | low
> 0.1    | 0.96  | high
> 0.2    | 1     | low
> 0.2    | 1     | high
> ```
>
> ```
> Num Workers: 50
> Margin | Power | Variance
> 0.05   | 0.925 | low
> 0.05   | 0.61  | high
> 0.1    | 1     | low
> 0.1    | 1     | high
> 0.2    | 1     | low
> 0.2    | 1     | high
> ```

---

### Meta-Review · Area_Chair_oQBY · 2023-09-16

**Recommendation:** 5

**Metareview:**

This paper proposed a new dialogue task on a commercial video game, and conducted user study to analyze and evaluate the behavior of LLM-augmented dialogues and game-designers' writing.

All the reviewers agree that this paper is both sound and exciting. But they also all have some concerns around the task evaluation and dataset analyses.

The discussions around the study size (especially around the wording of "large-scale") during rebuttal and analysis of the statistical power should be included in the camera-ready. It would also be good if the authors could discuss how this specific task would shed light on general dialogue research beyond games.

---

### Decision · Program_Chairs · 2023-10-07

**Decision:**

Accept-Findings

**Comment:**

This paper proposed a new dialogue task on a commercial video game, and conducted user study to analyze and evaluate the behavior of LLM-augmented dialogues and game-designers' writing.

All the reviewers agree that this paper is both sound and exciting. But they also all have some concerns around the task evaluation and dataset analyses.

The discussions around the study size (especially around the wording of "large-scale") during rebuttal and analysis of the statistical power should be included in the camera-ready. It would also be good if the authors could discuss how this specific task would shed light on general dialogue research beyond games.